microbiology/ecology/molecular biology

16S rRNA gene, *Calidris*, diversity, gastrointestinal tract, microbiota

**Author for correspondence:**
Kirsten Grond
e-mail: kirsten.grond@uconn.edu

†Present address: Department of Biomedical Sciences, Loyola University Chicago, Maywood, IL, USA.

# Spatial heterogeneity of the shorebird gastrointestinal microbiome

Kirsten Grond, Hannah Guilani† and Sarah M. Hird

Department of Molecular and Cell Biology, University of Connecticut, Storrs, CT, USA

 KG, 0000-0001-6197-7139; SMH, 0000-0002-1998-9387

The gastrointestinal tract (GIT) consists of connected structures that vary in function and physiology, and different GIT sections potentially provide different habitats for microorganisms. Birds possess unique GIT structures, including the oesophagus, proventriculus, gizzard, small intestine, caeca and large intestine. To understand birds as hosts of microbial ecosystems, we characterized the microbial communities in six sections of the GIT of two shorebird species, the Dunlin and Semipalmated Sandpiper, identified potential host species effects on the GIT microbiome and used microbial source tracking to determine microbial origin throughout the GIT. The upper three GIT sections had higher alpha diversity and genus richness compared to the lower sections, and microbial communities in the upper GIT showed no clustering. The proventriculus and gizzard microbiomes primarily originated from upstream sections, while the majority of the large intestine microbiome originated from the caeca. The heterogeneity of the GIT sections shown in our study urges caution in equating data from faeces or a single GIT component to the entire GIT microbiome but confirms that ecologically similar species may share many attributes in GIT microbiomes.

## 1. Introduction

The gastrointestinal tract (GIT) consists of connected structures that are involved in different aspects of digestion, energy metabolism, immunity and endocrinology [1–3]. The variety of functions, structures and physiologies of GIT sections potentially provide different habitats for microorganisms, which collectively form the GIT microbiome. To understand vertebrates as hosts of microbial ecosystems, a detailed description of these different habitats is essential.

The avian GIT is the structure that connects the bill to the cloaca, and includes the oesophagus, crop (functions in food storage), proventriculus (chemical digestion), gizzard (physical

digestion), small intestine (nutrient absorption), caeca (fermentation), large intestine (nutrient and water absorption) and cloaca (terminus of digestive, reproductive and urinary tracts). Several factors, such as pH, oxygen concentration and availability of nutrients, can potentially influence microbial communities along the avian GIT. For example, the pH in the GIT decreases from the oesophagus to the gizzard, where mechanical digestion of food particles occurs. In vultures, the extremely low pH in the stomach was hypothesized to result in selective filtering of the incoming microbial community, to retain a microbial community specialized to digest their scavenger diet [4].

After lagging behind mammalian microbiome research [5,6], avian microbiome studies are finally rising in numbers. However, our knowledge of the diversity and distribution of microbes in the GIT of most bird species is still virtually non-existent. A thorough mapping of microbiomes within bird hosts would allow for estimating gamma diversity as well as documenting intra and inter-individual variation in GIT microbiome. Insight into spatial distribution of the GIT microbiome within hosts is also essential to inform future microbiome sampling, and help narrow the place of origin and niche specialties for bacterial species.

Here, we sample six distinct GIT sections (oesophagus, proventriculus, gizzard, small intestine, caeca, large intestine) from adults of two species of shorebird, the Dunlin (*Calidris alpina*) and Semipalmated Sandpiper (*Calidris pusilla*). These shorebirds were collected during the two to three weeks in May that they stage in Delaware Bay on their way to the Arctic breeding grounds. During this period, they double in body mass [7] while foraging in mixed species flocks on a near exclusive diet of horseshoe crab (*Limulus polyphemus*) eggs [8]. Investigating microbiome composition in different sections of the GIT from two host species experiencing similar environmental and dietary conditions provides a relatively well-controlled *in situ* comparison of the two species and the six GIT sections. To elucidate the microbial ecology and spatial biogeography of the shorebird GIT microbiome, we (i) characterize the microbial communities in different sections of the shorebird GIT, (ii) identify potential effects of host species on the GIT microbiome, and (iii) use microbial source tracking to determine microbial origin throughout the GIT.

# 2. Methods

## 2.1. Sampling

Shorebirds used in our study consisted of capture fatalities in Delaware Bay, and were obtained from the Delaware Museum of Natural History, Wilmington, DE. No birds were euthanized for our study. All carcasses belonged to adult birds and were collected on the same day in May 2018 within 5 min of death, and were frozen at −20°C for four weeks prior to dissection.

We dissected out the whole GIT from six Semipalmated Sandpipers and six Dunlin after tying off the oesophagus and cloaca with cotton string to retain all GIT content and prevent outside contamination. We cleaned the exterior of the GIT by rinsing it with a 10% bleach solution prior to sampling individual GIT sections. GITs were separated into six sections (figure 3): oesophagus, proventriculus, gizzard, small intestine, caeca and large intestine. Tissue and content from the large intestine and oesophagus that touches the cotton string was discarded to prevent contamination. Hereafter, we defined the 'upper GIT' to include the first three sections (oesophagus, proventriculus, gizzard), and the 'lower GIT' to include the posterior three sections (small intestine, caeca, large intestine).

## 2.2. Extraction and sequencing

DNA was extracted using the Qiagen PowerFecal extraction kit (Qiagen, Hilden, Germany) following manufacturer's protocols with the exception of DNA being diluted in 50 µl instead of 100 µl in the final step. The oesophagus, proventriculus, caeca and large intestine were added to the extraction bead tube. For the gizzard, its content and a subsection of the gizzard internal wall was added to the bead tube, and for the small intestine, a two-inch section from the middle of the organ was added to the extraction bead tube. The V4 region of the 16S rRNA gene was sequenced at the UConn Microbial Analysis, Resources, and Services facility. The Quant-iT PicoGreen kit was used to quantify DNA concentrations, and 30 ng of extracted DNA was used as the template to amplify the V4 region of the 16S rRNA gene. V4 primers with Illumina adapters and dual barcodes were used for amplification (515F and 806R) [9,10]. PCR conditions consisted of 95°C for 3.5 min, 30 cycles of 30 s at 95.0°C, 30 s at 50.0°C and 90 s at 72.0°C, followed by final extension at 72.0°C for 10 min. PCR products were

normalized based on the concentration of DNA from 250 to 400 bp and pooled. Pooled PCR products were cleaned using the Mag-Bind RxnPure Plus (Omega Bio-tek) according to the manufacturer's protocol, and the cleaned pool was sequenced on the MiSeq using v2 2 × 250 base pair kit (Illumina, Inc., San Diego, CA). Additionally, three negative extraction controls and two PCR controls were sequenced.

## 2.3. Sequence analyses

The DADA2 (v. 1.12.1) pipeline [11] in R v. 3.6.0 [12] was used to process sequence data. DADA2 calls operational taxonomic units (OTUs) from sequence-based microbial communities by performing stringent quality control steps and subsequently calling each unique amplicon sequence variant (ASV) an OTU. After quality assessment, sequences were trimmed to remove low quality read areas, paired-end sequences were merged and chimeras removed. Sequences were assigned taxonomically using RDP's Naive Bayesian Classifier [13] with the Silva reference database (v. 128) [14]. Sequences identified as chloroplast and mitochondria were removed from the dataset. Likely sequence contaminants were identified and removed using the decontam package (v. 1.4.0) in R [15], which identified contaminant ASVs using the negative field control and PCR control. For phylogenetic analysis, a multiple sequence alignment was generated using the *DECIPHER* package (v. 1.12.1) in R [16], and a phylogenetic tree of all remaining ASVs was constructed with the *phangorn* package version 2.4.0 [17].

## 2.4. Statistical analyses

### 2.4.1. Alpha diversity

Two measures of alpha diversity were calculated: the observed number of ASVs and the Shannon diversity index [18], using the *phyloseq* package (v. 1.28.0) [19]. Samples were rarefied 10 times to a depth of our lowest sample (3637 sequences) prior to alpha diversity analysis, to confirm repeatability of results after rarefaction. Repeated measures analysis of variance (rANOVA) was used to determine whether alpha diversity of GIT sections differed from each other and among host species.

### 2.4.2. Beta diversity

Microbiome community analysis was conducted using the *phyloseq* package and results were visualized using the *ggplot2* package (v. 3.2.1) [19,20]. Relative abundances of bacterial taxa were calculated per GIT section for Dunlin and Semipalmated Sandpipers. To assess beta diversity, non-metric multidimensional scaling (NMDS) analysis was applied to Bray–Curtis, unweighted UniFrac and weighted UniFrac distances [21]. Beta dispersion of Bray–Curtis, weighted UniFrac and unweighted UniFrac distance matrices were statistically compared among all samples from different GIT sections and host species, using the *betadisper* and *permutest* function from the *vegan* package (v. 2.5.6) [22]. To determine which metadata variables (i.e. host species, GIT section, host sex) were correlated with microbiome composition, permutational multivariate analysis of variance (perMANOVA) was applied using the *adonis* function from the *vegan* package.

### 2.4.3. Community composition

We characterized the GIT microbiome of Dunlin and Semipalmated Sandpipers per GIT section using relative abundance of multiple taxonomic levels. We identified differential abundance of genera in different GIT section using the *DeSeq2* package (v. 1.24.0) in R [23]. *p*-Values were corrected with the Benjamini and Hochberg false discovery rate for multiple testing [24] and genera were identified as differentially abundant if the corrected *p*-values < 0.01.

### 2.4.4. Microbial source tracking and random forest modelling

To determine the potential origin of the bacteria found in GIT sections, we used the microbial source-tracking method FEAST in R [25]. FEAST uses an expectation-maximization based method that estimates which fraction of the microbial community in the input microbial community was derived from which potential source environment. Data were structured following the multiple sinks protocol

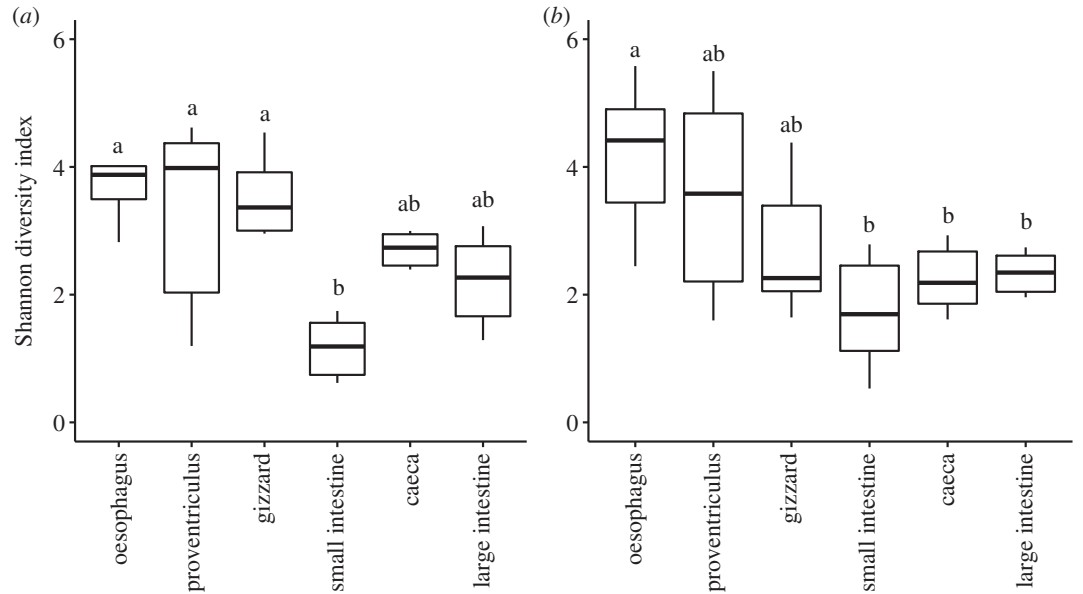

**Figure 1.** Alpha diversity (Shannon Diversity Index) of gut microbiomes of Dunlin (*a*) and Semipalmated Sandpipers (*b*), divided by six gut sections: oesophagus, proventriculus, gizzard, small intestine, large intestine and caeca. Letters represent significance at $\alpha = 0.05$.

from https://github.com/cozygene/FEAST. Per bird, each GIT section was identified as a sink, starting with the large intestine, with all anterior sections as sources.

To test whether microbiomes can predict host species and GIT section of origin, random forest classification and regression models [26] were applied using packages *randomForest*, *plyr*, *rfUtilities* and *caret* in R [27–30]. The 5% lowest abundance ASVs were removed and data was Z-transformed prior to performing classification and regression analyses to allow comparison across samples. Both analyses were run with 10 000 random trees, and regression analysis was run with 10 000 permutations. As our dataset was insufficient to generate a test set, random forest models were conducted with 71-fold cross-validation, withholding one sample per cross-validation.

## 3. Results

After quality control, we retained 2 586 814 high-quality 16S rRNA gene sequences in 71 samples with an average of 36 434 ± 2406 s.e. sequences/sample. One sample from the small intestine of the Dunlin (sample ID: D2SB02) did not amplify or sequence and was removed from further analyses.

### 3.1. Alpha diversity

Alpha diversity results did not differ among 10 rarefactions performed (ANOVA: $F_{1,708} = 0$, $p = 0.990$). Shannon's alpha diversity did not differ between host species (ANOVA: $F_{1,69} = 0.829$, $p = 0.366$) and sexes (ANOVA: $F_{1,69} = 0.05$, $p = 0.944$). Overall, alpha diversity significantly differed among GIT sections in both Dunlin (rANOVA: $F_{5,20} = 5.74$, $p = 0.002$) and Semipalmated Sandpipers (rANOVA: $F_{5,25} = 5.82$, $p < 0.001$). Alpha diversity was significantly different between the small intestine and the upper GIT organs in Dunlin (adj. $p = 0.001$–0.012; figure 1). In Semipalmated Sandpipers, the lower GIT organs differed significantly in alpha diversity from the oesophagus (adj. $p = 0.005$–0.04), but not from the proventriculus and gizzard (adj. $p = 0.07$–0.99; see table 1 for all results).

### 3.2. Beta diversity

We visually detected clustering in samples from the large intestine and caeca, and to a lesser extent in samples collected from the small intestine (figure 2). Samples collected from the upper GIT clustered together and did not cluster by GIT section. Beta dispersion significantly differed among GIT sections using weighted ($F_{5,65} = 2.58$, $p = 0.04$) and unweighted UniFrac ($F_{5,65} = 15.59$, $p \leq 0.001$) distances, but

**Table 1.** TukeyHSD Pairwise comparison of alpha diversity (Shannon) among GIT sections in Dunlin and Semipalmated Sandpipers. Shannon's diversity estimates averages and standard error (s.e.) per gut section are shown in parentheses after section. Significance was assigned at adjusted $p < 0.05$. Significant results are italicized.

| | Dunlin | | | Semipalmated Sandpiper | | |
| --- | --- | --- | --- | --- | --- | --- |
| | Shannon's diversity | | | Shannon's diversity | | |
| | section 1 ± s.e. \| section 2 ± s.e. | | $p$ | section 1 ± s.e. \| section 2 ± s.e. | | $p$ |
| oesophagus—proventriculus | 3.88 ± 0.34 \| 3.28 ± 0.63 | | 0.868 | 4.17 ± 0.8 \| 3.54 ± 0.67 | | 0.907 |
| oesophagus—gizzard | 3.88 ± 0.34 \| 3.29 ± 0.42 | | 0.890 | 4.17 ± 0.8 \| 2.71 ± 0.44 | | 0.201 |
| oesophagus—small intestine | 3.88 ± 0.34 \| 1.17 ± 0.20 | | *0.001* | 4.17 ± 0.8 \| 1.72 ± 0.37 | | *0.005* |
| oesophagus—caeca | 3.88 ± 0.34 \| 2.57 ± 0.22 | | 0.183 | 4.17 ± 0.8 \| 2.25 ± 0.22 | | *0.042* |
| oesophagus—large intestine | 3.88 ± 0.34 \| 2.21 ± 0.29 | | *0.046* | 4.17 ± 0.8 \| 2.16 ± 0.28 | | *0.030* |
| proventriculus—gizzard | 3.28 ± 0.63 \| 3.29 ± 0.42 | | 1.000 | 3.54 ± 0.67 \| 2.71 ± 0.44 | | 0.759 |
| proventriculus—small intestine | 3.28 ± 0.63 \| 1.17 ± 0.20 | | *0.010* | 3.54 ± 0.67 \| 1.72 ± 0.37 | | 0.061 |
| proventriculus—caeca | 3.28 ± 0.63 \| 2.57 ± 0.22 | | 0.785 | 3.54 ± 0.67 \| 2.25 ± 0.22 | | 0.314 |
| proventriculus—large intestine | 3.28 ± 0.63 \| 2.21 ± 0.29 | | 0.388 | 3.54 ± 0.67 \| 2.16 ± 0.28 | | 0.248 |
| gizzard—small intestine | 3.29 ± 0.42 \| 1.17 ± 0.20 | | *0.009* | 2.71 ± 0.44 \| 1.72 ± 0.37 | | 0.597 |
| gizzard—caeca | 3.29 ± 0.42 \| 2.57 ± 0.22 | | 0.755 | 2.71 ± 0.44 \| 2.25 ± 0.22 | | 0.973 |
| gizzard—large intestine | 3.29 ± 0.42 \| 2.21 ± 0.29 | | 0.358 | 2.71 ± 0.44 \| 2.16 ± 0.28 | | 0.943 |
| small intestine—caeca | 1.17 ± 0.20 \| 2.57 ± 0.22 | | 0.168 | 1.72 ± 0.37 \| 2.25 ± 0.22 | | 0.954 |
| small intestine—large intestine | 1.17 ± 0.20 \| 2.21 ± 0.29 | | 0.462 | 1.72 ± 0.37 \| 2.16 ± 0.28 | | 0.979 |
| caeca—large intestine | 2.57 ± 0.22 \| 2.21 ± 0.29 | | 0.984 | 2.25 ± 0.22 \| 2.16 ± 0.28 | | 1.00 |

did not differ using Bray–Curtis distances ($F_{5,65} = 1.12$, $p = 0.36$). Beta dispersion did not differ among Dunlin and Semipalmated Sandpipers (Bray–Curtis: $F_{1,69} = 1.15$, $p = 0.31$; weighted UniFrac: $F_{1,69} = 0$, $p = 0.99$; unweighted UniFrac: $F_{1,69} = 0.79$, $p = 0.36$).

All GIT sections differed significantly from each other regardless of distance matrix applied in both Dunlin and Semipalmated Sandpipers (table 2; PerMANOVA: $R^2 = 0.26–0.40$, $p < 0.001$). Host species microbiomes were only significantly different from each other when using Bray–Curtis distance matrices, and explained only 3% of variation in microbiome compositions (PerMANOVA: $R^2 = 0.03$, $p < 0.002$). We did not detect differences in microbiome composition between male and female birds in either species ($p = 0.051–0.549$)

## 3.3. Community composition

Overall, the dominant phylum detected across most GIT sections and both bird species was Proteobacteria, followed by Firmicutes and Fusobacteria (figure 3). Combining data for both species, the upper GIT contained a total of 345 genera as opposed to 164 genera in the lower GIT, of which 133 genera were shared among the upper and lower GIT. The oesophagus contained the most genera, followed by the next two posterior sections: the proventriculus and gizzard (figure 4). A majority of genera were unique to the oesophagus in both Dunlin ($n = 125$) and Semipalmated Sandpipers ($n = 71$), followed by genera that were detected in all sections of the upper GIT. Thirty-two and 24 genera were detected in all sections of the GIT in Dunlin and Semipalmated Sandpipers, respectively (figure 4).

In both shorebird hosts, we detected an increase in Bacteroidia classes after the small intestine, which was accompanied by a drop in Sphingobacteria. The *Bacteriodes* genus included a majority of the Bacteroidia sequences (68.5%). Within the Deferribacteres class, all sequences belonged to *Mucispirillum schaedleri*, which is the only species in the Deferribacteres phylum [31]. A majority of differentially abundant genera belonged to the Proteobacteria phylum (figure 5). The oesophagus and proventriculus differed the least in significantly different genera ($n = 11$), followed by the proventriculus and gizzard ($n = 17$). The largest number of genera were differentially abundant in the small intestine compared to the gizzard ($n = 66$). The caeca contained nine genera that were

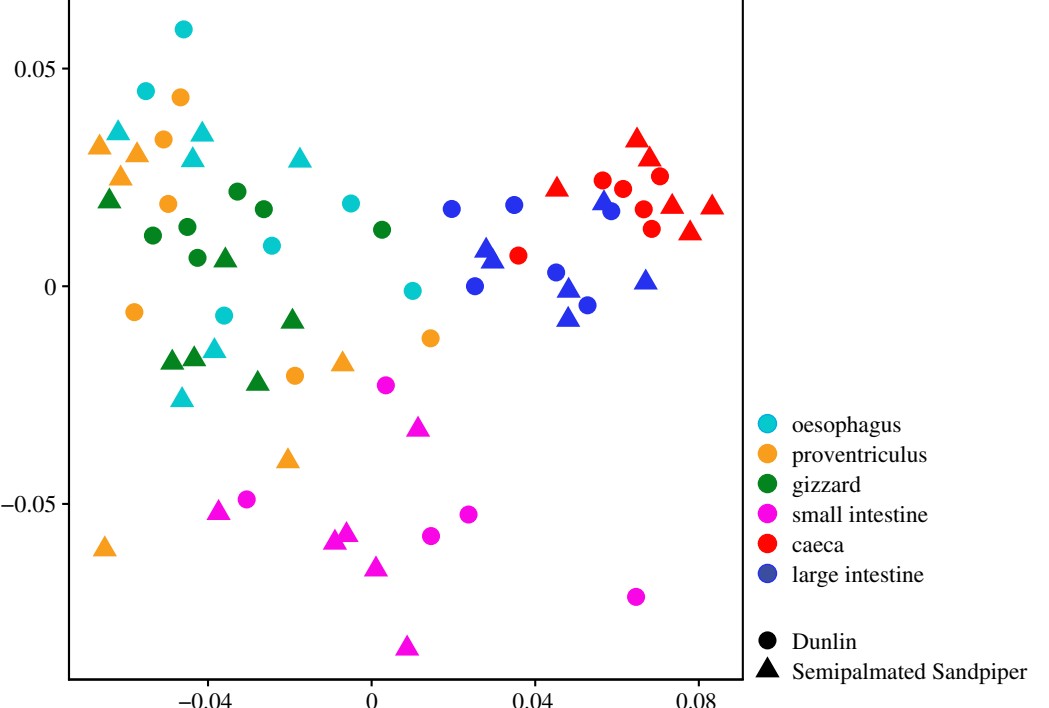

**Figure 2.** Non-metric Multidimensional Scaling ordination constructed from Bray–Curtis distance matrix of gut microbiomes in different gut sections collected from Dunlin and Semipalmated Sandpipers. Shapes represent host species, and colours represent gut section.

**Table 2.** PerMANOVA (adonis) tests for relative contribution and significance of three factors to variation in Bray–Curtis, and weighted and unweighted UniFrac Distance Matrices constructed from gut microbiomes of Dunlin and Semipalmated Sandpipers. Results are shown as $R^2/p$-value.

| | Dunlin | | | Semipalmated Sandpiper | | |
|---|---|---|---|---|---|---|
| | Bray | W. UniFrac | U. UniFrac | Bray | W. UniFrac | U. UniFrac |
| species[a] | 0.03/0.002 | 0.02/0.092 | 0.02/0.086 | 0.03/0.002 | 0.02/0.092 | 0.02/0.086 |
| GI section | 0.27/0.001 | 0.33/0.001 | 0.40/0.001 | 0.26/0.001 | 0.35/0.001 | 0.38/0.001 |
| sex | 0.04/0.104 | 0.05/0.128 | 0.03/0.549 | 0.04/0.051 | 0.03/0.362 | 0.03/0.311 |

[a]Test statistics are the same for both species.

significantly higher in abundance compared to the adjacent sections: the small intestine and the large intestine (table 3). In the other sections, only the genus *Anaerobispirillum* had a higher abundance in the gizzard than in the adjacent sections (figure 5).

The caeca and large intestine showed an increase in Fusobacteria, Deferribacteres and Bacteroidetes compared to the anterior sections, especially the small intestine, in both hosts (figure 3). All sequences within the Fusobacteria belonged to the class *Fusobacteriia*; 87.2% of *Fusobacteriia* were classified as the genus *Fusobacterium*, and the remaining 12.8% of sequences belonged to the *Cetobacterium* genus. The Bacteroidetes phylum consisted of five classes, and was dominated by the *Bacteroidia* (79.9%) and the *Flavobacteriia* (15.6%) classes (figure 6).

Dunlin had lower relative abundances of Firmicutes in their gizzards and small intestines (30.2%) compared to Semipalmated Sandpipers (43.2%), but showed higher abundances of Fusobacteria in the caeca and large intestines (21.4%) than Semipalmated Sandpipers (12.6%).

## 3.4. Microbial source tracking and random forest modelling

In both Dunlin and Semipalmated Sandpipers, microbial source tracking revealed that the large intestine microbiome was predominantly sourced from the caecal microbiome, followed by the small

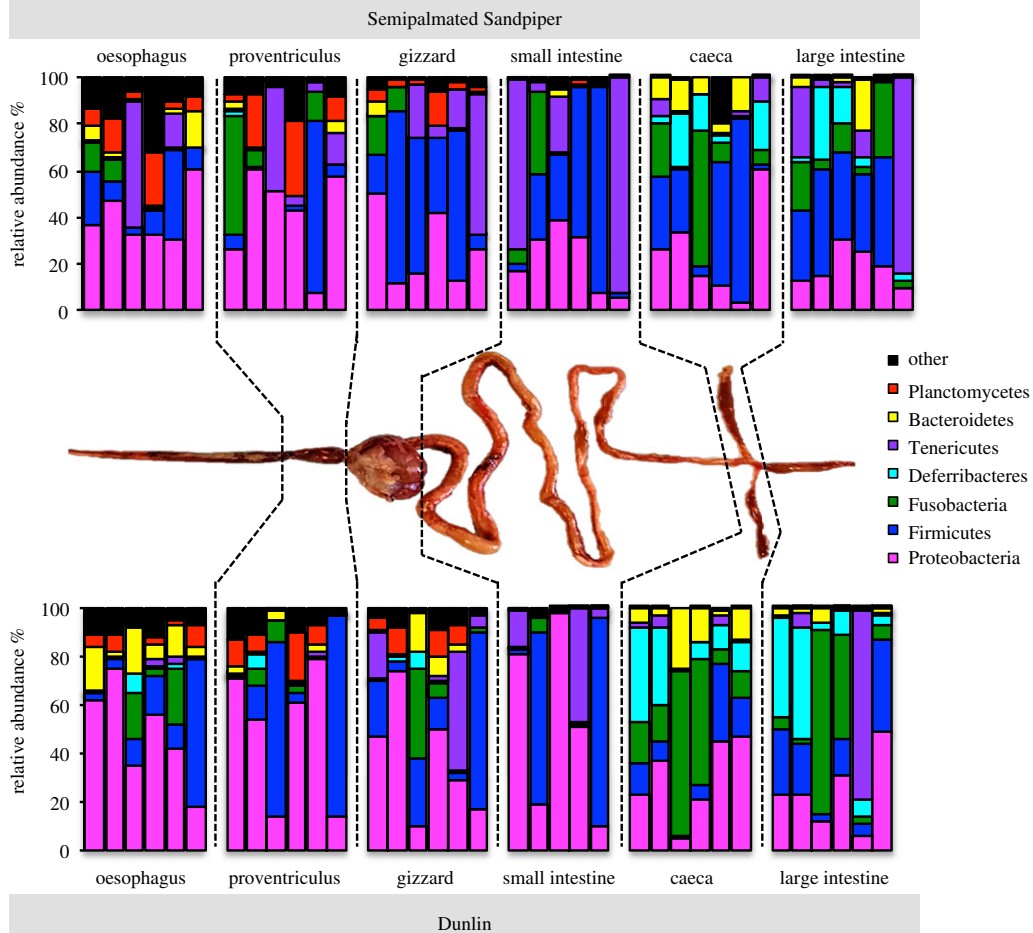

**Figure 3.** Phylum relative abundance for different sections of the Dunlin and Semipalmated Sandpiper guts. Bars represent six different individuals per species, with the exception of the Dunlin small intestine. Individuals are displayed in the same order in each GIT section.

intestine (figure 7). The caecal microbiome did now show clear microbiome sourcing from any anterior GIT sections (unknown source = 95.4%), indicating a unique microbiome within the caeca. In most hosts, the gizzard microbiome was sourced from the anterior GIT sections: the proventriculus and oesophagus.

Random Forest modelling did not confidently assign our samples to the correct host species or GIT section. The average out-of-the-bag (OOB) error validating our model was 19.7%, and the OOB for the regression analysis of the mean prediction of individual decision trees was 54.9%, both with 22 variables tried at each split. Classification OOB error in our regression model ranged from 27.3% for the small intestine to 91.7% for the proventriculus.

# 4. Discussion

The gastrointestinal tract of a host includes many microhabitats that contain distinct and diverse microbial communities. The variation in the communities across individual hosts is poorly understood and not quantified for most species. Here we have described the microbial communities in six distinct segments of the GIT in two wild shorebird species under extremely similar environmental and ecological conditions to understand the biogeography of the shorebird microbiome. We detected similar patterns in diversity and community structure among different sections of the GIT in the Dunlin and Semipalmated Sandpiper. The three upper GIT sections showed higher diversity than the lower sections, and differed in their microbiome composition. Higher alpha diversity in the upper GIT is likely due to the influx of a larger diversity of microorganisms that are associated with environment and diet.

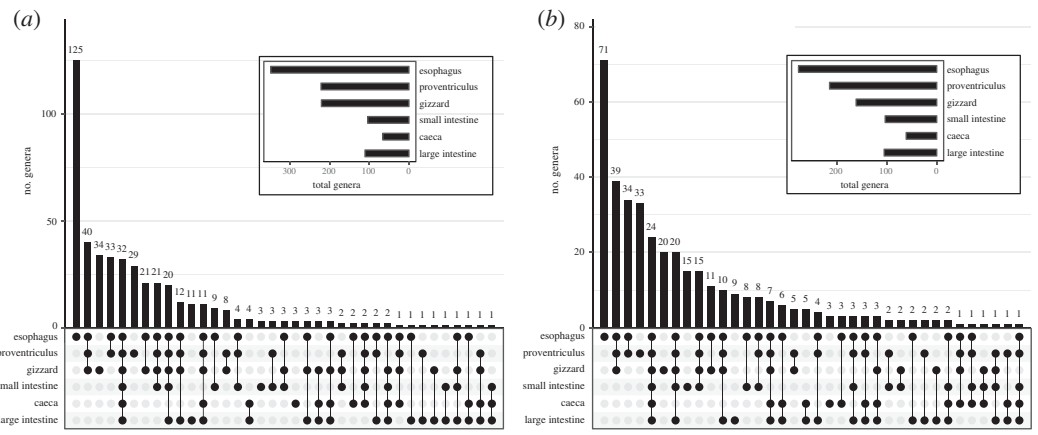

**Figure 4.** Shared genera between GIT sections in Dunlin (*a*) and Semipalmated Sandpipers (*b*). Numbers above the bars represent genera shared among the GIT sections identified with dots. Total number of genera found in each GIT section is shown in the box.

Our alpha diversity results are similar to those reported from other bird species. For example, higher bacterial diversity and distinct microbial communities are found in the proventriculus and gizzard of Taihu geese (*Anser cygnoides* spp.) [32]. Contrary to our results, Canada geese (*Branta canadensis*) harboured a higher bacterial diversity in their lower than upper GIT [33]. However, we did detect a similar low diversity in the small intestine in our study as in the duodenum of Canada geese [33] and in the ileum of Ostriches (*Struthio camelus*) [34]. The small intestine is thought to be the gut section that supports the least microorganisms, due to its high concentration of enzymes [35] and the low pH in the anterior section: the gizzard [36].

Dissimilar to our results, Japanese Quail (*Coturnix japonica*) have highest bacterial diversity in the caeca, and no clear differences among other GI sections with regards to diversity [37]. However, in Ostriches the ileum contained a different microbial community than the other GIT sections they investigated [34], which matches our results for the small intestine in shorebirds.

A majority of avian studies that investigate different GIT sections were conducted in broiler chickens. Chicken caeca are markedly different from other GIT sections, but, similar to studies above, no other sections appeared to contain distinct microbiomes [38]. However, chicken data may be difficult to extrapolate to wild birds, due to conditions of captivity differing from natural environments and the limited genetic variation among broiler chickens.

## 4.1. Host filtering

The ecological filtering model posits that microbes in the gut are selectively filtered from the environment based on host traits that they are adapted to [39]. Selective passage of microbes through the GIT has been documented in a number of host taxa, including bumblebees (*Bombus terrestris*) [40], vultures [4], juveniles of freshwater fish species [41] and a number of mammalian species [42]. Stomach acids in humans with a pH of 1–3.5 are able to degrade nucleic acids [43], and the same is likely for birds, which have a pH of 1–3 in their stomachs [44]. Therefore, our detection of decreased alpha diversity and community complexity in the lower GIT could be the result of host filtering of bacteria in the upper GI sections.

We observed a microbial trickle-down effect from the oesophagus to the proventriculus, gizzard and small intestine, which could indicate a decreasing influence of environmental and dietary microbes on the GIT microbiome. Although no studies have addressed this topic, physiological changes in oxygen concentration and pH along the GIT likely limit the passage of environmental microorganisms [44,45]. There appeared to be a separation in microbial sources between the proventriculus, gizzard, and small intestine, and the caeca and large intestine. Interestingly, the caecal microbiome did not show substantial sourcing from any of the anterior GIT sections, but provided a substantial portion of the large intestine microbiome. Previously, the caeca were shown to have microbiomes different from other sections of the GIT [38,46–48]. In our study, the caeca microbiome differed from all GIT sections, except from the large intestine. Similarities between the caeca and large intestine are not surprising, as there is bi-directional flow of digesta between these two organs [49]. The lack of microbial connectivity with the anterior sections could indicate that the caecal microbiome could be established

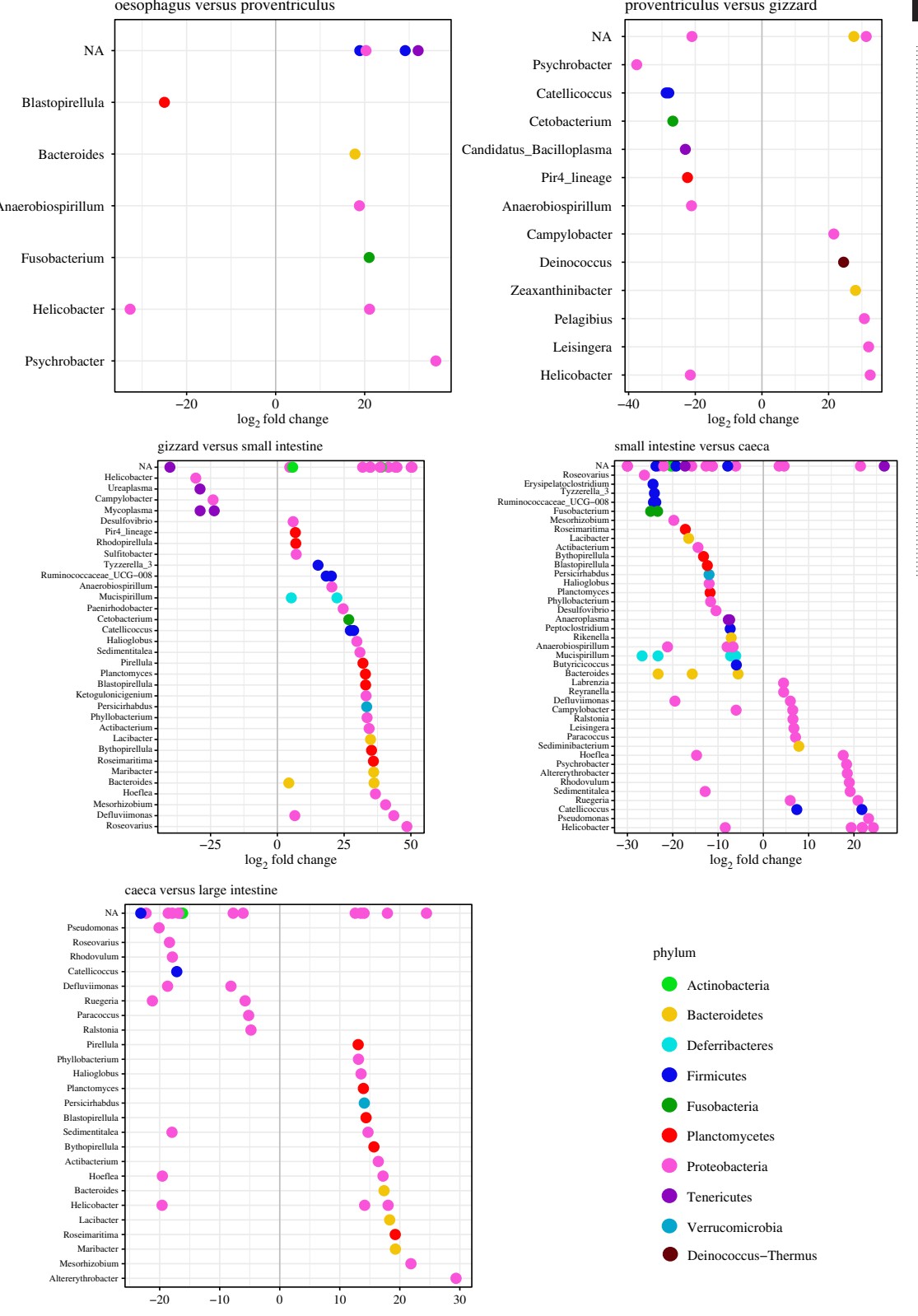

**Figure 5.** Significant differentially abundant genera among adjacent sections of the GIT from the oesophagus towards the large intestine. A positive $log_2$ fold change indicates higher relative abundance of genera first section mentioned in the title, and a negative $log_2$ fold change represents genera with higher relative abundance in the second section mentioned in the title. NA includes sequences that could not be confidently classified to genus level. Colours represent different phyla the displayed genera are part of. Genera were considered differentially abundance if the adjusted $p$-value < 0.01.

**Table 3.** Significantly differentially abundant bacterial genera in GIT sections of shorebirds. Genera mentioned were significantly higher than compared to their two adjacent sections. Significance was assigned if adjusted $p$-values < 0.01.

| GIT section | significant differentially abundant genera | |
| --- | --- | --- |
| | count | genus |
| proventriculus | 0 | |
| gizzard | 1 | *Anaerobiospirillum* (Proteobacteria) |
| small intestine | 0 | |
| caeca | 9 | *Planctomycetes* (Planctomycetes) |
| | | *Persicirhabdus* (Verrucomicrobia) |
| | | *Blastopirellula* (Planctomycetes) |
| | | *Bythopirellula* (Planctomycetes) |
| | | *Bacteroides* (3 ASVs; Bacteroidetes) |
| | | *Lacibacter* (Bacteroidetes) |
| | | *Roseimaritima* (Planctomycetes) |
| | | *Mesorhizobium* (Proteobacteria) |
| | | *Altererythrobacter* (Proteobacteria) |

prior to our sampling and is stable over age. An alternative, non-exclusive explanation of our findings could be that host filtering in shorebirds is especially rigid in the caeca, which would prevent recruitment from incoming microorganisms.

## 4.2. Interspecific differences

We did not detect significant differences in microbiome composition and diversity indices between Dunlin and Semipalmated Sandpipers. These species co-occur on the beaches in Delaware Bay, DE, and rely nearly exclusively on diets of horseshoe crab eggs during this staging period [8]. The similarity in environment and ecology should reveal species-specific microbial associations, and the lack thereof implies similar processes occurring in both hosts. This result is not unexpected as both Dunlin and Semipalmated Sandpipers belong to the genus *Calidris* and share most life-history characteristics, such as breeding area, migration flyway and parental care system [50,51]. However, we are limited in our conclusions by our sample size, which includes six GIT samples from six individuals per species. It is possible that we currently lack the resolution to reveal more subtle differences in microbiomes among hosts that are driven by underlying traits. A phylogenetically informed comparative study or experimental manipulation of captive animals would better elucidate the role of host genetics on the microbiome.

## 4.3. Microbial function in the shorebird microbiome

In birds, the caeca and large intestine are involved in fermentation of dietary compounds, electrolyte and water reabsorption, and nutritional uptake [6]. The bacterial communities of the caeca and large intestine were markedly different from the anterior sections with respect to relative increases in the Deferribacteres and Bacteroidetes phyla. Within the Deferribacteres phylum, the *Mucispirillum* genus consisted solely of *Mucispirillum schaedleri* [31], and Deferribacteres were largely absent in upstream GIT sections. *Mucispirillum schaedleri* is a known colonizer of the mucus layer in the mammalian GIT [52–54], and has been previously detected in chickens [55,56], vampire finches (*Geospiza septentrionalis*; Michel *et al.* [57]) and turkeys [58]. In chickens, caecal *M. schaedleri* was significantly positively associated with fat deposition [55]. At time of sampling, Dunlin and Semipalmated Sandpipers were rapidly gaining weight and depositing fat in preparation for migration, indicating a potentially similar role for *M. schaedleri* in shorebirds.

Bacteroides are among the most common microorganisms detected in animal GITs [59,60]. Their metabolism is mainly based on degradation of dietary and mucus glycoproteins, which are known to play a role in immune system regulation [61]. In chicken caeca, Bacteroides were the main microbes

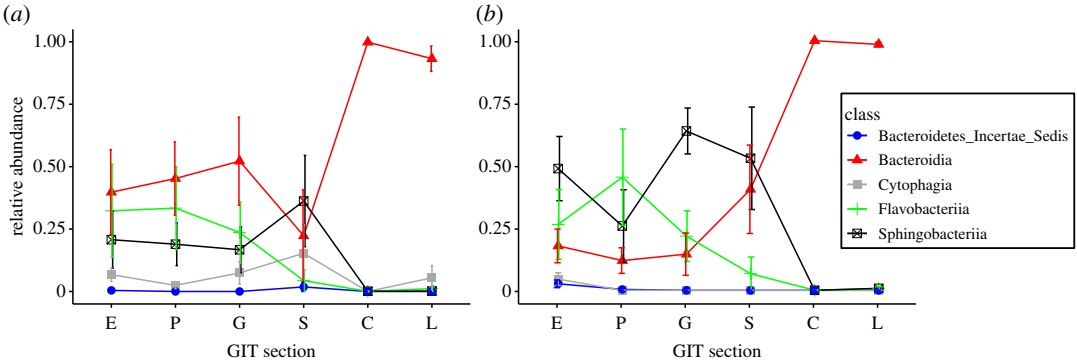

**Figure 6.** Relative abundances per GIT section of classes within the Bacteroidetes phylum in Dunlin (*a*) and Semipalmated Sandpipers (*b*). GIT section is depicted in order of ingestion through the oesophagus (E), proventriculus (P), gizzard (G), small intestine (S), caeca (C) and large intestine (L).

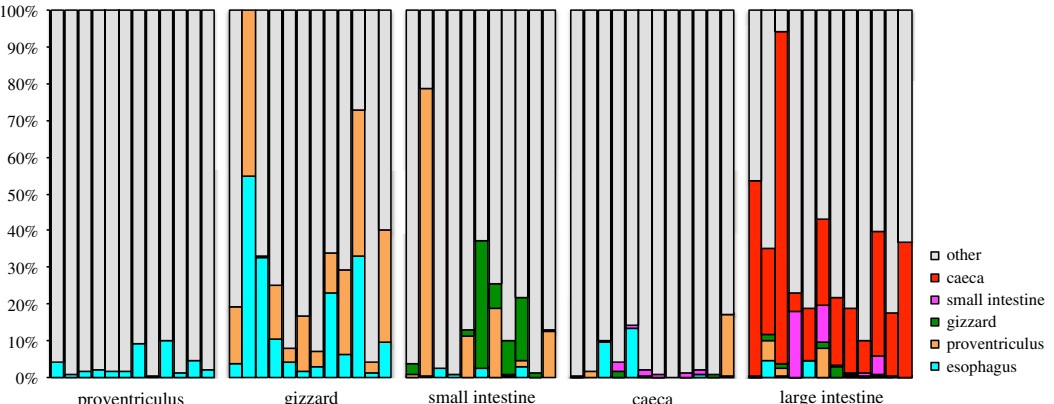

**Figure 7.** Relative contributions of GIT sections to their posterior section. The first six bars per section represent Dunlin, and the second six bars per section represent samples from Semipalmated Sandpipers.

involved in the degradation of non-starch polysaccharides and monosaccharides [62,63]; a function that could translate to shorebird caeca.

# 5. Conclusion

We detected immense variation in the microbiomes of six sites of the GIT in two species of shorebird. Faecal samples are often used as a proxy for gut microbiomes, and show the most similarity to the microbiome of the large intestine [34]. Although faecal samples are unlikely to capture the entire GIT microbiome community and variety, it is often the only non-invasive method available for investigating the microbiomes of wild animals. Therefore, we advise authors that use faecal samples to clearly define which microbiome their samples represent. Our results show that the microbiomes of the upper and lower GIT differ markedly in diversity, community composition and sources but that adjacent organs are less distinct. The heterogeneity of the GIT sections shown in our study urges caution in equating data from faeces or a single GIT component to the entire GIT microbiome but confirms that ecologically similar species may share many attributes in GIT microbiomes.

Ethics. Samples were collected with permission from the Delaware Division of Fish and Wildlife-Department of Natural Resources and Environmental Control (2018-WSC-031 to K.G.), and the Federal Bird Banding Permit (23332 to Delaware Division of Fish and Wildlife-Department of Natural Resources and Environmental Control).
Data accessibility. Sequences and metadata are available at Figshare (www.figshare.com/articles/Spatial_heterogeneity_of_the_shorebird_gut_microbiome/9792668), and sequences are also available at the NCBI SRA under BioProject ID: PRJNA580479. R scripts for analyses are deposited on Github at: https://github.com/KCGrond/shorebird_gut_heterogeneity.git.

Authors' contributions. K.G. conceived of the study, designed the study, collected field data, conducted data analysis and drafted the manuscript; H.G. and K.G. carried out the molecular laboratory work. S.M.H. participated in the design of the study and critically revised the manuscript; H.G. participated in the design of the study and helped draft the manuscript. All authors gave final approval for publication and agree to be held accountable for the work performed therein.

Competing interests. The authors declare no competing interests.

Funding. This work was supported by the University of Connecticut (start-up funding to S.M.H.). Fieldwork of this project was funded, in part, through a grant from the United States Fish & Wildlife Service's State Wildlife Grant Program. This work does not represent the opinions of the State of Delaware, Delaware Department of Natural Resources & Environmental Control or Delaware Division of Fish & Wildlife.

Acknowledgements. We thank the Delaware Shorebird Project and the Delaware Museum of Natural History for providing specimens, specifically Jean Woods, Audrey DeRose-Wilson, Nigel Clark, Jacquie Clark and Gregory Breese. We thank Inglis E. Tucker for help with dissections. We thank the associate editor, Dr Tezel, and two anonymous reviewers for their comments, which improved our manuscript. Our work was conducted on the traditional and unceded land and territories of the Lenape and Nanticoke Nations (DE), and the Mohegan, Mashantucket Pequot, Eastern Pequot, Golden Hill Paugussett, and Nipmuc Peoples (CT).

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
