## [Reviewer comments · Royal Society Open Science]

Review History

RSOS-191609.R0 (Original submission)

Review form: Reviewer 1

Is the manuscript scientifically sound in its present form?

Yes

Are the interpretations and conclusions justified by the results?

Yes

Is the language acceptable?

Yes

Do you have any ethical concerns with this paper?

No

Have you any concerns about statistical analyses in this paper?

Yes

Recommendation?

Accept with minor revision (please list in comments)

Comments to the Author(s)

Major comments:

Overall this paper is well organized, the methods and analyses are sound, and the results are interesting. I think it will be an important contribution to our field.

My only major concerns are as follows:

- 1) Some of the statistical analyses would benefit from adjustments to control for repeated measures from a single individual
- 2) Data does not seem to be available at provided link, and no link is provided for a script that would allow people to replicate your analyses

Minor comments:

ABSTRACT

23 - alpha and genus diversity? This reads strangely; maybe taxonomic and alpha diversity, or rephrase another way.

26 - The language of "sourcing" and "originating" without the context of your cool analyses is confusing. In the abstract before reading the paper it seems as though it refers to your sampling. Consider add a sentence about the method, or use different language for the abstract.

INTRO

54 - "our knowledge of the diversity and distribution of microbes in the GIT of most bird species"

63 - can you estimate the age of the birds? or just specify juvenile or adult.

METHODS

81 - what kind of string was used?

Overall, great job on this section. All of it was very clear and tight.

RESULTS

179 - Did you use a repeated measures ANOVA to compare samples only within individuals? As your samples show considerable variation among individuals, it may be beneficial to account for this statistically.

180 - sentence order is off, I think you meant to have "Shannon's alpha diversity" and "host species" switched

248 - "small intestine" does not need to be capitalized

345 - "have been shown to show" consider rephrasing for less redundancy. Also, consider adding citations to this sentence.

DISCUSSION

312 - typo : "have share"

FIGURES

fig. 1 - consider adding full sample type name, vertically or at an angle on the axis for ease of interpretation

table 1 - typo - "proventriculus" is missing a "t"

Review form: Reviewer 2

Is the manuscript scientifically sound in its present form?

Yes

Are the interpretations and conclusions justified by the results?

Yes

Is the language acceptable?

Yes

Do you have any ethical concerns with this paper?

Yes

Have you any concerns about statistical analyses in this paper?

No

Recommendation?

Accept with minor revision (please list in comments)

Comments to the Author(s)

Grond et al. have evaluated the microbiome of six sections of the GIT in six individuals of two species of shorebirds. This is a good study and I enjoyed reading it. It is useful for the avian microbiome community. It is well written with adequate amount of information in the methods and the results. The figures are nice and well presented. The analyses are appropriate. I don't have any major comments but I hope that my minor comments will be useful to the authors during the revision of the paper.

Minor comments:

- Abstract and discussion: In general, I believe the term "gut microbiome" is most often used to refer to the microbiota of the large intestine. Meaning that researchers sample feces in order to draw conclusions about the large intestinal microbiome, but the typical word used is often gut microbiome. I don't think anyone believes that the bacterial community will be similar throughout the entire GIT (especially since this term includes everything from the bill to the cloaca). It has also been shown in previous studies that this is indeed the case. I think the authors are correct in stating that the GIT sections are different (which they have evaluated) and that fecal samples do not portray everything in the GIT (which they have not evaluated). But I also want to urge the authors that it might be useful to be slightly careful with this wording, because no one seriously believes you will get an accurate picture of the esophagus microbiome by sampling feces. So the point of feces not representing the entire GIT community becomes a bit meaningless. The point of fecal sampling is not to measure the entire GIT but to evaluate the large intestine non-invasively.
- L50: The cited paper has not studied the gizzard microbial community or its pH. Please revise.
- L53: I think "decades" is a bit exaggerated. Probably "years" would fit better. Mammalian microbiome research in its current form is also relatively new.
- Methods sampling: It is my understanding that one needs a permit for trapping birds and another permit for collecting birds. Feel free to correct me if I'm wrong. Even if the birds used in this study were accidentally killed during the trapping procedure, don't the authors agree it would be appropriate to state the permit or licenses used for trapping and collecting, since this allowed the authors (or collaborators) to catch the birds in the first place?
- L105: As far as I'm aware, there is no MiSeq v4 kit. There are v2 and v3. Probably just a typo.

- L126: Dada2, vegan, phyloseq, DeSeq2, FEAST versions have not been specified. Since these kinds of software often make substantial changes between versions, it would be good to state version used for reproducibility reasons.
- L170 Data availability: Royal Society Data Policy states that “Datasets and code should be deposited in an appropriate, recognized, publicly available repository. Where no data-specific repository exists, authors should deposit their datasets in a general repository such as Dryad or Figshare.” First of all, I cannot find the sequences or the metatable of this study in the provided Figshare link. Regardless, I believe the 16S sequences in this study should be deposited in appropriate sequence databases such as SRA or ENA or DDBJ to allow for future re-analyses and meta-analyses. Figshare is not an appropriate repository for open sequence data. The metatable can be store on Figshare (in addition to the sequence repository).
- Table 1: Misspellings. Please check.
- Table 1: Curious as to why only p-values are provided in Table 1? Where are the effect sizes? p-values only tell the reader whether the test was significant or not at an arbitrary threshold. As a reader you want to see the results of the analysis. Is the esophagus more diverse than the gizzard? That’s not possible to tell from p-values. Please add to the table diversity values, so the reader will at least know which direction the difference is. Consider also stats from the ANOVA test.
- Figure 2: I like the colors used. They are easy to tell apart.
- L252: The word microbiome is used but I think the authors mean gastro-intestinal tract.
- L284: “Decreased alpha diversity and community complexity in the lower GIT could be the result of host filtering of bacteria in the upper GI sections.” Can the authors please explain this further how they mean? If the host kills certain bacteria in the upper GI sections with pH, the dead bacteria would still be present in the lower gut community as well due to the downward flow of content. This also does not explain why the diversity of bacteria is higher upper in the gut? Do the authors mean that the higher diversity in the upper gut is most likely derived from the diet and environmentally sourced bacteria?
- Discussion: I don’t know the word limit of this journal but if possible, I would love to read a little more extended discussion. The authors very briefly touch upon some of the interesting results they found, but there is very little integration on what it means and any comparisons with previous studies. There are a lot of similar studies that have been conducted in grouse, ostriches and sparrows for example.
- Overall, I think the manuscript is well-written and easy to read.

Decision letter (RSOS-191609.R0)

22-Oct-2019

Dear Dr Grond,

On behalf of the Editors, I am pleased to inform you that your Manuscript RSOS-191609 entitled "Spatial Heterogeneity of the Shorebird Gut Microbiome" has been accepted for publication in Royal Society Open Science subject to minor revision in accordance with the referee suggestions. Please find the referees' comments at the end of this email.

The reviewers and handling editors have recommended publication, but also suggest some minor revisions to your manuscript. Therefore, I invite you to respond to the comments and revise your manuscript.

- Ethics statement

- Data accessibility

<http://datadryad.org/submit?journalID=RSOS&manu=RSOS-191609>

- Competing interests

- Authors' contributions

- Acknowledgements

- Funding statement

Please ensure you have prepared your revision in accordance with the guidance at <https://royalsociety.org/journals/authors/author-guidelines/> -- please note that we cannot publish your manuscript without the end statements. We have included a screenshot example of

the end statements for reference. If you feel that a given heading is not relevant to your paper, please nevertheless include the heading and explicitly state that it is not relevant to your work.

Because the schedule for publication is very tight, it is a condition of publication that you submit the revised version of your manuscript before 31-Oct-2019. Please note that the revision deadline will expire at 00.00am on this date. If you do not think you will be able to meet this date please let me know immediately.

Please note that Royal Society Open Science charge article processing charges for all new submissions that are accepted for publication. Charges will also apply to papers transferred to Royal Society Open Science from other Royal Society Publishing journals, as well as papers

submitted as part of our collaboration with the Royal Society of Chemistry (<http://rsos.royalsocietypublishing.org/chemistry>).

Kind regards,
Lianne Parkhouse
Editorial Coordinator
Royal Society Open Science
openscience@royalsociety.org

on behalf of Dr Ulas Tezel (Associate Editor) and Kevin Padian (Subject Editor)
openscience@royalsociety.org

Associate Editor Comments to Author (Dr Ulas Tezel):

Dear Dr. Kirsten Grond:

Please see below the comments and suggested MINOR revisions made by the individual(s) who reviewed your manuscript. I would like to take your attention to one critical issue raised by the reviewers:

1. The link provided in the manuscript (10.6084/m9.figshare.9792668) is not valid, thus data provided in the link is not accessible. Please provide a valid link or update the link so the data can be easily accessed.

Reviewer comments to Author:

Reviewer: 1
Comments to the Author(s)

Major comments:

Overall this paper is well organized, the methods and analyses are sound, and the results are interesting. I think it will be an important contribution to our field.

My only major concerns are as follows:

- 1) Some of the statistical analyses would benefit from adjustments to control for repeated measures from a single individual
- 2) Data does not seem to be available at provided link, and no link is provided for a script that would allow people to replicate your analyses

Minor comments:

ABSTRACT

23 - alpha and genus diversity? This reads strangely; maybe taxonomic and alpha diversity, or rephrase another way.

26 - The language of "sourcing" and "originating" without the context of your cool analyses is confusing. In the abstract before reading the paper it seems as though it refers to your sampling. Consider add a sentence about the method, or use different language for the abstract.

INTRO

54 - "our knowledge of the diversity and distribution of microbes in the GIT of most bird species"

63 - can you estimate the age of the birds? or just specify juvenile or adult.

METHODS

81 - what kind of string was used?

Overall, great job on this section. All of it was very clear and tight.

RESULTS

179 - Did you use a repeated measures ANOVA to compare samples only within individuals? As your samples show considerable variation among individuals, it may be beneficial to account for this statistically.

180 - sentence order is off, I think you meant to have "Shannon's alpha diversity" and "host species" switched

248 - "small intestine" does not need to be capitalized

345 - "have been shown to show" consider rephrasing for less redundancy. Also, consider adding citations to this sentence.

DISCUSSION

312 - typo : "have share"

FIGURES

fig. 1 - consider adding full sample type name, vertically or at an angle on the axis for ease of interpretation

table 1 - typo - "proventriculus" is missing a "t"

Reviewer: 2

Comments to the Author(s)

Grond et al. have evaluated the microbiome of six sections of the GIT in six individuals of two species of shorebirds. This is a good study and I enjoyed reading it. It is useful for the avian microbiome community. It is well written with adequate amount of information in the methods and the results. The figures are nice and well presented. The analyses are appropriate. I don't have any major comments but I hope that my minor comments will be useful to the authors during the revision of the paper.

Minor comments:

- Abstract and discussion: In general, I believe the term "gut microbiome" is most often used to refer to the microbiota of the large intestine. Meaning that researchers sample feces in order to draw conclusions about the large intestinal microbiome, but the typical word used is often gut microbiome. I don't think anyone believes that the bacterial community will be similar

throughout the entire GIT (especially since this term includes everything from the bill to the cloaca). It has also been shown in previous studies that this is indeed the case. I think the authors are correct in stating that the GIT sections are different (which they have evaluated) and that fecal samples do not portray everything in the GIT (which they have not evaluated). But I also want to urge the authors that it might be useful to be slightly careful with this wording, because no one seriously believes you will get an accurate picture of the esophagus microbiome by sampling feces. So the point of feces not representing the entire GIT community becomes a bit meaningless. The point of fecal sampling is not to measure the entire GIT but to evaluate the large intestine non-invasively.

- L50: The cited paper has not studied the gizzard microbial community or its pH. Please revise.
- L53: I think “decades” is a bit exaggerated. Probably “years” would fit better. Mammalian microbiome research in its current form is also relatively new.
- Methods sampling: It is my understanding that one needs a permit for trapping birds and another permit for collecting birds. Feel free to correct me if I’m wrong. Even if the birds used in this study were accidentally killed during the trapping procedure, don’t the authors agree it would be appropriate to state the permit or licenses used for trapping and collecting, since this allowed the authors (or collaborators) to catch the birds in the first place?
- L105: As far as I’m aware, there is no MiSeq v4 kit. There are v2 and v3. Probably just a typo.
- L126: Dada2, vegan, phyloseq, DeSeq2, FEAST versions have not been specified. Since these kinds of software often make substantial changes between versions, it would be good to state version used for reproducibility reasons.
- L170 Data availability: Royal Society Data Policy states that “Datasets and code should be deposited in an appropriate, recognized, publicly available repository. Where no data-specific repository exists, authors should deposit their datasets in a general repository such as Dryad or Figshare.” First of all, I cannot find the sequences or the metatable of this study in the provided Figshare link. Regardless, I believe the 16S sequences in this study should be deposited in appropriate sequence databases such as SRA or ENA or DDBJ to allow for future re-analyses and meta-analyses. Figshare is not an appropriate repository for open sequence data. The metatable can be store on Figshare (in addition to the sequence repository).
- Table 1: Misspellings. Please check.
- Table 1: Curious as to why only p-values are provided in Table 1? Where are the effect sizes? p-values only tell the reader whether the test was significant or not at an arbitrary threshold. As a reader you want to see the results of the analysis. Is the esophagus more diverse than the gizzard? That’s not possible to tell from p-values. Please add to the table diversity values, so the reader will at least know which direction the difference is. Consider also stats from the ANOVA test.
- Figure 2: I like the colors used. They are easy to tell apart.
- L252: The word microbiome is used but I think the authors mean gastro-intestinal tract.
- L284: “Decreased alpha diversity and community complexity in the lower GIT could be the result of host filtering of bacteria in the upper GI sections.” Can the authors please explain this further how they mean? If the host kills certain bacteria in the upper GI sections with pH, the dead bacteria would still be present in the lower gut community as well due to the downward flow of content. This also does not explain why the diversity of bacteria is higher upper in the gut? Do the authors mean that the higher diversity in the upper gut is most likely derived from the diet and environmentally sourced bacteria?
- Discussion: I don’t know the word limit of this journal but if possible, I would love to read a little more extended discussion. The authors very briefly touch upon some of the interesting results they found, but there is very little integration on what it means and any comparisons with previous studies. There are a lot of similar studies that have been conducted in grouse, ostriches and sparrows for example.
- Overall, I think the manuscript is well-written and easy to read.

Author's Response to Decision Letter for (RSOS-191609.R0)

See Appendix A.

Decision letter (RSOS-191609.R1)

11-Nov-2019

Dear Dr Grond,

It is a pleasure to accept your manuscript entitled "Spatial Heterogeneity of the Shorebird Gastrointestinal Microbiome" in its current form for publication in Royal Society Open Science.

Before we proceed to the production stage, we ask that you please now make sure that your figshare article is finalised as a collection: <https://knowledge.figshare.com/articles/item/how-to-use-collections>

This will ensure that a formal DOI can be assigned to your dataset; this will enhance both visibility of your data, and ensure that your dataset can be cited appropriately with the DOI given. Once you have done this, please send an updated copy of your manuscript file (a clean Word document, with the updated data accessibility statement and figshare DOI) by return email.

Once we receive this, you can then expect to receive a proof of your article in the near future. Please contact the editorial office (openscience_proofs@royalsociety.org) and the production office (openscience@royalsociety.org) to let us know if you are likely to be away from e-mail contact -- if you are going to be away, please nominate a co-author (if available) to manage the proofing process, and ensure they are copied into your email to the journal.

Kind regards,
Lianne Parkhouse
Editorial Coordinator
Royal Society Open Science
openscience@royalsociety.org

on behalf of Dr Ulas Tezel (Associate Editor) and Kevin Padian (Subject Editor)
openscience@royalsociety.org

Appendix A

Dear editors,

Below we addressed the comments of Dr. Tezel and the two reviewers. We hope we have adequately addressed their concerns, and have modified our manuscript to be satisfactory for publication. We thank Dr. Tezel and the reviewers for their constructive comments and compliments.

Sincerely,

Kirsten Grond

- Ethics statement

>> We added an ethics section at the end of the manuscript with all permits and approvals.

- Data accessibility

>> the data accessibility section is updated

Associate Editor Comments to Author (Dr Ulas Tezel):

Dear Dr. Kirsten Grond:

Please see below the comments and suggested MINOR revisions made by the individual(s) who reviewed your manuscript. I would like to take your attention to one critical issue raised by the reviewers:

1. The link provided in the manuscript (10.6084/m9.figshare.9792668) is not valid, thus data provided in the link is not accessible. Please provide a valid link or update the link so the data can be easily accessed.

>> Our apologies for the broken link. We updated the link for figshare, and added the SRA BioProject number for NCBI.

Reviewer comments to Author:

Reviewer: 1

Comments to the Author(s)

Major comments:

Overall this paper is well organized, the methods and analyses are sound, and the results are interesting. I think it will be an important contribution to our field.

My only major concerns are as follows:

1) Some of the statistical analyses would benefit from adjustments to control for repeated measures from a single individual

>> We conducted a repeated measures anova to control for repeated measures from the same individual, but the results did not change the significance of our results ($p= 0.001-0.002$). We added in the use of rANOVA's for alpha diversity statistics to the methods and results.

2) Data does not seem to be available at provided link, and no link is provided for a script that would allow people to replicate your analyses

>> We updated the figshare link (our apologies!) and added the sequence data to the NCBI SRA. We also uploaded our R scripts to Github and added the link to our data accessibility statement.

Minor comments:

ABSTRACT

23 - alpha and genus diversity? This reads strangely; maybe taxonomic and alpha diversity, or rephrase another way.

>> We rephrased the sentence to read 'alpha diversity and genus richness', which was our intended meaning.

26 - The language of "sourcing" and "originating" without the context of your cool analyses is confusing. In the abstract before reading the paper it seems as though it refers to your sampling. Consider add a sentence about the method, or use different language for the abstract.

>> We modified the sentence to read: 'primarily originated from' instead of ' were primarily sourced from' to clarify our meaning.

INTRO

54 - "our knowledge of the diversity and distribution of microbes in the GIT of most bird species"

>> We edited our sentence to reflect the reviewers corrections

63 - can you estimate the age of the birds? or just specify juvenile or adult.

>> all birds used were adults, which we have now specified in the introduction and methods section.

METHODS

81 - what kind of string was used?

>> we used regular cotton string, and did not use the tissue and content directly next to the string. We added this to the methods.

Overall, great job on this section. All of it was very clear and tight.

RESULTS

179 - Did you use a repeated measures ANOVA to compare samples only within individuals? As your samples show considerable variation among individuals, it may be beneficial to account for this statistically.

>> see above

180 - sentence order is off, I think you meant to have "Shannon's alpha diversity" and "host species" switched

>> We thank the reviewer for noticing this mistake, and have rewritten the sentence.

248 - "small intestine" does not need to be capitalized

>> We removed the capitalization of small intestine.

345 - "have been shown to show" consider rephrasing for less redundancy. Also, consider adding citations to this sentence.

>> We replaced 'have been shown to show' with 'show', and added a citation by Videlvall and al 2018.

DISCUSSION

312 - typo : "have share"

>> We removed 'have' in this sentence.

FIGURES

fig. 1 - consider adding full sample type name, vertically or at an angle on the axis for ease of interpretation

>> We added the full sample type name to figure 1.

table 1 - typo - "proventriculus" is missing a "t"

>> We corrected this mistake.

Reviewer: 2

Comments to the Author(s)

Grond et al. have evaluated the microbiome of six sections of the GIT in six individuals of two species of shorebirds. This is a good study and I enjoyed reading it. It is useful for the avian microbiome community. It is well written with adequate amount of information in the methods and the results. The figures are nice and well presented. The analyses are appropriate. I don't

have any major comments but I hope that my minor comments will be useful to the authors during the revision of the paper.

Minor comments:

- Abstract and discussion: In general, I believe the term “gut microbiome” is most often used to refer to the microbiota of the large intestine. Meaning that researchers sample feces in order to draw conclusions about the large intestinal microbiome, but the typical word used is often gut microbiome. I don’t think anyone believes that the bacterial community will be similar throughout the entire GIT (especially since this term includes everything from the bill to the cloaca). It has also been shown in previous studies that this is indeed the case. I think the authors are correct in stating that the GIT sections are different (which they have evaluated) and that fecal samples do not portray everything in the GIT (which they have not evaluated). But I also want to urge the authors that it might be useful to be slightly careful with this wording, because no one seriously believes you will get an accurate picture of the esophagus microbiome by sampling feces. So the point of feces not representing the entire GIT community becomes a bit meaningless. The point of fecal sampling is not to measure the entire GIT but to evaluate the large intestine non-invasively.

>> We have changed gut microbiome to GIT microbiome throughout the paper. Technically, the gut microbiome represents the entire GIT, but with its use in current literature we agree with the author that it could be a confusing term.

Although we agree with the author that the comparison of esophagus and fecal microbiomes are meaningless, we do want to emphasize in our paper that data from feces or single GIT sections do not represent the GIT microbiome. Although gut microbiome is often used interchangeably with the large intestinal microbiome, this is rarely mentioned in publications. I have seen a number of publications equating fecal or even cloacal samples to represent the gut, with no specification of what the gut represents in their paper.

However, I have used fecal sampling in a number of my own studies and I agree it is the only non-invasive method available for evaluating the large intestine microbiome. I added a sentence to the conclusion section to clarify this:

“Although fecal samples are unlikely to capture the entire GIT microbiome community and variety, it is often the only non-invasive method available for investigating the microbiome of wild animals. Therefore, we advise authors that use fecal samples to clearly define which microbiome their samples represent.”

- L50: The cited paper has not studied the gizzard microbial community or it’s pH. Please revise.

>> Our apologies for the mistake. We changed the sentence to reflect the hypothesis of the authors that the acidic stomach causes the microbial filtering from face to intestine.

- L53: I think “decades” is a bit exaggerated. Probably “years” would fit better. Mammalian microbiome research in its current form is also relatively new.

>> We removed ‘several decades’, which changed the sentence to: “After lagging behind mammalian microbiome research”

- Methods sampling: It is my understanding that one needs a permit for trapping birds and another permit for collecting birds. Feel free to correct me if I'm wrong. Even if the birds used in this study were accidentally killed during the trapping procedure, don't the authors agree it would be appropriate to state the permit or licenses used for trapping and collecting, since this allowed the authors (or collaborators) to catch the birds in the first place?

>> We agree with Reviewer 2 and we added our permit numbers for trapping and collecting to the new required Ethics section.

- L105: As far as I'm aware, there is no MiSeq v4 kit. There are v2 and v3. Probably just a typo.

>> This is indeed a typo, our apologies. We used the v2 kit and changed this in the manuscript.

- L126: Dada2, vegan, phyloseq, DeSeq2, FEAST versions have not been specified. Since these kinds of software often make substantial changes between versions, it would be good to state version used for reproducibility reasons.

>> We agree with reviewer 2, and added the version numbers of the programs and packages used, with the exception of FEAST. We mistakenly identified FEAST as a package, when it is custom code described in the citation provided.

- L170 Data availability: Royal Society Data Policy states that "Datasets and code should be deposited in an appropriate, recognized, publicly available repository. Where no data-specific repository exists, authors should deposit their datasets in a general repository such as Dryad or Figshare." First of all, I cannot find the sequences or the metadata of this study in the provided Figshare link. Regardless, I believe the 16S sequences in this study should be deposited in appropriate sequence databases such as SRA or ENA or DDBJ to allow for future re-analyses and meta-analyses. Figshare is not an appropriate repository for open sequence data. The metadata can be stored on Figshare (in addition to the sequence repository).

>> we updated the figshare link and provided the SRA bioproject number that contains the raw sequences. Our apologies for the broken link.

- Table 1: Misspellings. Please check.

>> We corrected the misspelled proventriculus.

- Table 1: Curious as to why only p-values are provided in Table 1? Where are the effect sizes? p-values only tell the reader whether the test was significant or not at an arbitrary threshold. As a reader you want to see the results of the analysis. Is the esophagus more diverse than the gizzard? That's not possible to tell from p-values. Please add to the table diversity values, so the reader will at least know which direction the difference is. Consider also stats from the ANOVA test.

>> We added the diversity values to the table for clarification. Since we performed a TukeyHSD test we did not have test statistics to add for this.

- Figure 2: I like the colors used. They are easy to tell apart.

>> Thank you!

- L252: The word microbiome is used but I think the authors mean gastro-intestinal tract.

>> We meant the microbiome in general as a collection of microbial communities, but since our paper focuses on the GIT we realize this causes confusion. We changed microbiome to gastrointestinal tract for clarity.

- L284: “Decreased alpha diversity and community complexity in the lower GIT could be the result of host filtering of bacteria in the upper GI sections.” Can the authors please explain this further how they mean? If the host kills certain bacteria in the upper GI sections with pH, the dead bacteria would still be present in the lower gut community as well due to the downward flow of content.

>> In humans, nucleic acids were shown to be digested in the stomach with a pH of 1.3-3.5 (Liu et al. 2016. Scientific Reports). Birds have a pH from 1-3 (Beasley et al. 2015. PloS One). We therefore believe that the avian proventriculus and gizzard play a role in host filtering by not only killing bacteria, but also (partially) degrading their DNA. We added this information to our paper.

This also does not explain why the diversity of bacteria is higher upper in the gut? Do the authors mean that the higher diversity in the upper gut is most likely derived from the diet and environmentally sourced bacteria?

>> We indeed mean that the upper GIT has a wider microbial exposure to environment and diet, which likely results in the higher alpha diversity. We added a sentence to clarify this.

“Higher alpha diversity in the upper GIT is likely due to the influx of a larger diversity of microorganisms that are associated with environment and diet.”

- Discussion: I don’t know the word limit of this journal but if possible, I would love to read a little more extended discussion. The authors very briefly touch upon some of the interesting results they found, but there is very little integration on what it means and any comparisons with previous studies. There are a lot of similar studies that have been conducted in grouse, ostriches and sparrows for example.

>> We have added a couple sections to the discussion further explaining our results(see highlighted sections. We also added data from Videvall et al. concerning ostrich microbiomes, but our comparative ability was limited as they did not investigate microbiomes of the upper GIT. We searched for the grouse and sparrow GIT section publications mentioned by the reviewer but were unable to find any relevant papers unfortunately.

- Overall, I think the manuscript is well-written and easy to read.

>> Thank you!